# Transcriptional Analysis Reveals the Differences in Response of Floral Buds to Boron Deficiency Between Two Contrasting *Brassica napus* Varieties

**DOI:** 10.3390/plants14060859

**Published:** 2025-03-10

**Authors:** Zhexuan Jiang, Lan Liu, Sheliang Wang, Xiangsheng Ye, Zhaojun Liu, Fangsen Xu

**Affiliations:** 1National Key Laboratory of Crop Genetic Improvement, Microelement Research Center, Huazhong Agricultural University, Wuhan 430070, China; jiangzhexuan@webmail.hzau.edu.cn (Z.J.);; 2Key Laboratory of Genome Research and Genetic Improvement of Xinjiang Characteristic Fruits and Vegetables, Institute of Horticultural Crops, Xinjiang Academy of Agricultural Sciences, Urumqi 830091, China

**Keywords:** boron deficiency, *Brassica napus*, buds, reproductive development, transcriptome, boron transporters, phytohormone

## Abstract

Boron (B) is an essential micronutrient for the development of crops, and its reproductive stage is particularly sensitive to B deficiency. *Brassica napus* L., as an important oil-crop species, is extremely vulnerable to B deficiency. The typical B-deficient symptom of “flowering without seed setting” usually results in severe yield loss. However, few studies have focused on the response of the reproductive organs to B deficiency. In this study, the B-efficient variety “Zhongshuang 11” (ZS11) and the B-inefficient variety “Westar 10” (W10) of *Brassica napus* were selected to be cultivated at the developmental stage (BBCH15) in a pot experiment, both with and without B supply. Clear phenotype differences in B deficiency between the two varieties’ flowers appeared only at the reproductive stage, and only W10 showed symptoms of delayed flower opening, stigma exsertion, and resulted in abortion. Transcriptome analysis for the early buds of both varieties between B supply (+B) and free (−B) treatments revealed that W10 had more differentially expressed genes (DEGs) corresponding to its greater susceptibility to −B. As two potential mechanisms to improve B-efficient utilization, we focused on analyzing the expression profiles of B transporter-related genes and phytohormone metabolism-related genes. *BnaC05.NIP7;1*, *BnaC08.NIP3;1,* and *BnaBOR2s* were identified as the key genes which could enhance the capacity of B translocation to buds of ZS11. Additionally, combined with a phytohormone concentration measurement, we showed that a significant increase in IAA and a drastic decrease in JA could predominantly lead to the abnormal development of W10’s buds. *BnaC02.NIT2* (*Nitrilase 2*) and *BnaKAT5s* (*3-Ketoacyl-CoA Thiolase 5*), which are IAA and JA biosynthesis genes, respectively, could be the key genes responsible for the changes in IAA and JA concentrations in W10’s buds under −B. These candidate genes may regulate the genotype differences in the response of the rapeseed reproductive stage to −B between different B-efficient varieties. It also has potential to breed rapeseed varieties with B-efficient utilization in the reproductive stage, which would improve the seed yield under −B condition.

## 1. Introduction

Boron (B) is an essential micro-nutrient for plant growth and development [1,2]. The concentration of B in plants is in the range of 2~100 mg/kg, and 60% to 98% of total B in plants is stored in the cell wall [3,4]. Ample evidence has demonstrated that B is primarily involved in the cross-linking process of two apiose residues in the rhamnogalacturonan II (RG-II) in order to maintain the stability and extensible structure of the cell wall [4,5]. Therefore, B in plants was difficult to remobilize, and the symptoms of B deficiency first appeared in vigorous tissues, such as the root tip, young leaves, shoot apical meristems, and reproductive organs [2,6,7,8]. The demand for B increases when plants enter the reproductive stage. The importance of B for the development of reproductive organs was reported as early as 1930s [9], and afterwards, B deficiency-reduced fields were successively found in cotton [10], rice [11], and rapeseed [12]. It was found that B deficiency prevented anthers from dehiscing properly, and mature pollen grains became sunken, papilla cells were shriveled, the pollen tubes were unable to reach ovules, and ultimately, this resulted in a lower yield [13,14,15,16]. Moreover, foliar supplementation with B could significantly enhance pollen viability and in vitro culture. Adequate B could also promote pollen germination and pollen tube elongation [17,18]. However, the molecular mechanism causing B deficiency-induced reproductive abortion is still unclear.

To date, many bioinformatics analyses and gene function studies of plant’s response to B deficiency have been focused on the vegetative stages rather than the reproductive stage. Two potential mechanisms and some underlying genes enhancing B transport and utilization efficiency were discovered through dedicated efforts [19,20,21]. On the one hand, alterations in transcript abundance and protein accumulation levels of several members in NIPs (Nodulin26-like Intrinsic Protein) and BORs promoted B uptake and translocation in response to B deficiency [22,23,24]. *NIP5;1* was localized in the epidermal cells of the root tip and was involved in the uptake of B from the environment [25,26]. Then, B was transmitted into the xylem via BOR1/2 and loaded toward to shoot [22,27]. *NIP6;1* was expressed in nodes and transported B from the xylem to the phloem [28]. *NIP7;1*, *BnaA02.NIP6;1*, *BnaBOR2s,* and *BnaC04.BOR1* were expressed in reproductive organs, but their involvement in B transport remains unknown [14,29,30,31,32]. On the other hand, alterations in phytohormone concentration and cell wall-related gene expression also responded to B deficiency, which affected B utilization efficiency in plants [33,34,35]. In the root, B deficiency upregulated the expression of JA synthesis genes *AOCs* and increased the concentration of JA that inhibited root elongation; the mutant *jar1-1*, which diminished JA signaling in plants, could increase the biomass and the concentration of B under B deficiency [36,37]. Similarly, B deficiency-induced IAA synthesis, and the sensitivity of root cells to B deficiency could be effectively reduced in the IAA-responsive mutations *eir1* and *aux1* [38]. Plant reproductive growth is more sensitive to −B than vegetative growth, as the development of the plant flower organ requires more B. However, the molecular mechanism involved in the transport and distribution of B to reproductive organs, and the response of the reproductive growth to −B, depends on these two potential mechanisms, which are still uncovered.

Rapeseed (*Brassica napus* L.) is a prominent oil crop worldwide and is extremely sensitive to B deficiency. The availability of B is poor in many arable lands, which makes it a limiting factor for yield and quality [39,40]. Previous studies have made much progress in the response mechanism of *Brassuca napus* to −B. They found that the B-efficient variety ‘Qingyou 10’ (QY10) coped better with B deficiency than the B-inefficient variety ‘Westar 10’ (W10) [19]. Transcriptomic research of roots and leaves at the vegetative stage between these two rapeseed varieties revealed the differences in response to B deficiency, which involved the regulation of B transporter genes and phytohormone-related genes [20,34]. Nevertheless, limited attention has been paid to the differences in the response of the two varieties of rapeseed to −B at the reproductive stage. Recently, ‘Zhongshuang 11’ (ZS11), which was also a B-efficient variety, has been sequenced with a high-quality genome and can be utilized for transcriptome analysis [41]. Therefore, in this study, we conducted a pot experiment with ZS11 and W10, and then, we observed the differences in the morphology and physiology between the two rapeseed varieties at the reproductive stage with sufficient and deficient B supplies. Transcriptome sequencing was performed on floral buds from the pot experiment to reveal the differential expression of B transporter genes and phytohormone-related genes. This aimed to excavate candidate genes and provide a basis for studying the molecular mechanism involved in the response of reproductive growth to B deficiency in rapeseed.

## 2. Results

### 2.1. Distinct Phenotypical and Physiological Responses of Westar 10 and Zhongshuang 11 to Boron Deficiency at the Reproductive Stage

To assess the phenotypical differences in response to boron (B) deficiency between ‘Zhongshuang 11’ (ZS11) and ‘Westar 10’ (W10) at the reproductive stage, both varieties were first cultured for 25 d with five true leaves in Hoagland solution with normal B condition. After being washed with ultrapure water, the homogenous plants were then transplanted to pots with adequate (+B) or free (−B) B supplies. Before bolting, and no clear phenotypical differences were observed between the two varieties (Figure 1A). However, the B concentration in old leaves of both varieties was already significantly reduced to ~4 µg/g by −B (Figure 1B). In the case of B concentration in the root, no difference was observed in each variety. Interestingly, a significant B reduction was only observed in the young leaves of W10 but not in ZS11, suggesting that these two varieties had different sensitivity levels to −B (Figure 1B).

At the flowering stage, only the old leaves of ZS11 presented a significant reduction in B concentration among different tissues, including the stem, old leaves, new leaves, and floral organs, but the B concentration of all these tissues in W10 decreased significantly under −B (Figure 1C). A closer observation showed that −B tended to reduce the number of opening flowers in both genotypes. Specifically, W10 also demonstrated stigma exsertion, a typical phenomenon induced by −B (Appendix A and Figure 1D–G). We also investigated silique phenotypes and found that −B restricted silique elongation in both varieties (Figure 1H), and only W10 showed a significant reduction in silique length (Figure 1I). In addition, compared to ZS11, −B significantly reduced the seed number per silique, and ultimately led to poor yield (Figure 1J–L). Taken together, our results suggest that the reproductive organs of W10 were more sensitive to −B than ZS11.

### 2.2. Differentially Expressed Genes and Altered Biological Processes in Floral Buds of Westar10 and Zhongshuang11 Under Boron Deficiency

To investigate the −B altered transcriptome response of W10 and ZS11 at the early reproductive phase, we harvested early closed buds with similar sizes (buds with 2–4 mm length) under sufficient (+B) or deficient (−B) B conditions, and we performed RNA-seq analysis using the Illumina sequencing platform. A total of 6.79 × 108 raw reads and 6.66 × 108 clean reads were sequenced in our samples. The base error rate of all samples was 0.03%, and the average GC content was 44.95%. As the parameter of the reads quality, the average of Q20 (the percentage of the base alignment accuracy above 99%) or Q30 (the percentage of the base alignment accuracy above 99.9%) was 97.63% or 93.22%, respectively. Furthermore, the ratio of clean reads was successfully aligned with the reference genome for each sample, ranging from 89.6% to 91.06% (Appendix A). These overall data indicated that the sequencing was highly accurate and sufficient for the quantitative analysis of gene expression.

We first performed principal component analysis (PCA) for the transcriptome data and divided 12 samples into four groups, with each group representing ZS11 or W10 buds from +B or −B conditions. Although the three samples of ZS11 from the −B condition tended to have higher variation within the treatment, our PCA was not able to completely separate +B and −B samples of ZS11 (Figure 2A). In contrast, the samples of +B and −B treatments of W10 could be divided by PCA, and the samples with +B conditions were closely related to + or −B treated samples of ZS11, but the −B treated samples of W10 were more distant (Figure 2A). By using an adjusted *p* value ≤ 0.05 and an absolute value of log_2_Fold-change ≥ 1 as the standards to define differentially expressed genes (DEGs), we identified a total of 16,561 or 330 DEGs in buds of W10 or ZS11, and each with 7954/8607 (up-/down-regulated) or 281/49 (up-/down-regulated) genes, respectively (Figure 2B, Appendix A). These data indicated that W10’s buds responded more dramatically to −B than ZS11’s buds. A Venn analysis of DEGs showed that 84/14 DEGs were commonly up-/down-regulated and 48/7 DEGs presented the opposite, regulated by −B induced in W10 and ZS11’s buds. However, the majority of DEGs were specifically altered within each variety (Figure 2C). These results suggested that the buds of W10 and ZS11 had distinct responses to −B at the global transcriptome level.

To determine the functions of the DEGs in buds between W10 and ZS11 varieties under +B and −B conditions, we further performed a gene ontology (GO) enrichment analysis of the DEGs to distinguish the major biological process (BP), molecular function (MF), and cellular component (CC). A total of 504 or 228 GO terms were significantly enriched in W10 or ZS11, each with 289/57/158 or 133/14/81 in BP/CC/MF, respectively (Appendix A). Then, we listed the top 20 most significant BP terms and the top 10 most significant CC and MF terms according to −log_10_(*p*-value) in both lines (Figure 2D,E). In the case of W10 buds, the most frequently associated BP terms were plant-type secondary cell wall biogenesis, xylan biosynthetic process, and lignin biosynthetic process, which were closely relevant to the cell wall. We also found that carbohydrate transport, monosaccharide transmembrane transport, amino acid transport, cellular copper ion homeostasis, and the long-chain fatty acid metabolic process had been enriched in W10’s buds. Furthermore, cytosol and cysteine-type peptidase activities were mostly enriched in CC and MF terms (Figure 2D). In terms of ZS11, the cutin transport, fatty acid transport, cellular oxidant detoxification, borate transporter, and so on, were enriched in BP. The external side of the plasma membrane’s and the fatty acid transmembrane transporter’s activity were dominantly in CC and MF terms (Figure 2E). These results suggested that the regulated pathway induced by −B between the W10 and ZS11 buds had significant differences.

### 2.3. Expression Profiles of BnaAQPs and BnaBORs in Two Lines of Buds Responding to Boron Deficiency

More studies have demonstrated that many members of the AQPs/Aquaporins play crucial roles in facilitating B uptake and translocation, especially at the vegetative stages rather than the reproductive stage. Therefore, we aligned sequences of *AtAQPs* genes to the *Brassica napus* reference genome. A total of 132 *BnaAQPs* were globally identified, which included 30 *BnaNIPs*, 12 *BnaSIPs* (*Small and basic Intrinsic Protein*), 53 *BnaPIPs* (*Plasma membrane Intrinsic Protein*), and 37 *BnaTIPs* (*Tonoplast Intrinsic Protein*) (Appendix A). The expression profiling of these genes was visualized using FPKM (fragments per kilobase of transcript per million reads mapped) values. The results showed that most of the *BnaPIPs* and *BnaTIPs* genes were relatively higher than others in the buds of both varieties (Figure 3A). In addition, *BnaPIP1;2s* and *BnaPIP2;7s*, together with Bna*TIP1;1s*, *BnaTIP1;2s*, *BnaA05.SIP1;1*, and *BnaC07.NIP1;2* had higher relative expression levels in *BnaPIPs*, *BnaTIPs*, *BnaSIPs*, and *BnaNIPs*, respectively (Appendix A). It was implied that these genes played essential functions in the floral organs.

We further gathered the DEGs of *BnaAQPs* and focused on the normalized FPKM values using a heat map to better display the differential response to −B between W10 and ZS11’s buds (Figure 3B, Appendix A). We found that the expression profiles of these DEGs were briefly grouped into three categories. (1) Gene expression was inhibited by −B mainly in W10, which included three homologous genes of *BnaNIP6;1s* (*BnaC06.NIP6;1*, *BnaA07.NIP6;1* and *BnaA02.NIP6;1c*), *BnaC09.PIP2;4*, *BnaA04.SIP2;1*, *BnaC01.TIP2;1,* and so on. (2) Gene expression was induced by −B primarily in W10, which had three homologous genes of *BnaNIP5;1s* (*BnaA03.NIP5;1*, *BnaA02.NIP5;1* and *BnaC03.NIP5;1*), *BnaAnn.PIP2;7*, four homologous genes of *BnaTIP2;1s* (*BnaA01.TIP2;1*, *BnaA05.TIP2;1*, *BnaC05.TIP2;1* and *BnaC03.TIP2;1*), and so on. (3) Gene expression was induced by −B majority in ZS11, which only contains two genes, *BnaC08.NIP3;1* and *BnaC05.NIP7;1*. These two genes could play an important role in enhancing the capacity of B translocation in buds of B-efficient varieties.

Moreover, we also detected 21 *BnaBORs* members in the transcriptome data, including six *BnaBOR1s*, two *BnaBOR2s*, two *BnaBOR3s*, five *BnaBOR4s*, two *BnaBOR6s,* and four *BnaBOR7s* (Appendix A, Figure 3C). Interestingly, the general expression profiles of *BnaBORs* in both varieties were not affected by −B, except *BnaA01.BOR7*, which was significantly suppressed by −B in W10’s buds. Additionally, we found that two members of *BnaBOR2s* (*BnaC04.BOR2* and *BnaAnn.BOR2*) were highly expressed in buds, especially in the ZS11 variety, which indicated that *BnaBOR2s* may play an important role in the floral bud development under the −B condition.

### 2.4. Genes Involving IAA and JA Biosynthesis Pathways in Response to B Starvation in Floral Buds of Two Varieties

To investigate the regulation of different phytohormone-related genes in rapeseed buds under −B, we summarized the genes in terms of phytohormone synthesis, metabolism, and signaling in Brassica napus based on Liu [42]. This included 881 abscisic acid (ABA)-related genes, 872 ethylene (ETH)-related genes, 425 brassinosteroids (BR)-related genes, 466 gibberellins (GA)-related genes, 1312 auxin (IAA)-related genes, 895 jasmonate (JA)-related genes and 448 cytokinin (CK)-related genes (Appendix A). DEGs analysis was performed on ‘ZS11’ and ‘W10’ buds in response to −B, and the results showed that there were only nine hormone-related DEGs, excluding CK-related genes, in ZS11 varieties, and all of them were significantly induced by −B. However, 220 up-regulated and 190 down-regulated hormones-related DEGs were found in the bud of W10. The top three hormones with the highest number of DEGs were IAA, JA, and ABA (Figure 4A–C, Appendix A). Therefore, we hypothesized that IAA, JA, and ABA would be the major hormones involved in the response to −B in rapeseed buds. Then, we once again harvested unified similar-sized buds from the two varieties in pot experiments and measured the concentration of these three hormones (Figure 4D–F). The data indicated that the concentration of IAA was significantly increased in W10 by −B, and the concentration of JA was decreased by −B in both varieties, but the decreased degree in W10’s bud was much higher than ZS11. The concentration of ABA was not altered by −B in ZS11 and W10’s buds.

Further analysis of gene expression patterns in the IAA biosynthesis pathway was performed concerning the Trp/tryptophan-dependent pathway, which predominantly exists in plants [43]. We blasted a total of 60 genes in the *Brassica napus* genome involved in the tryptophan-dependent pathway and listed 32 genes with the FPKM values among all treatments (Appendix A). The expression pattern of these genes was visually demonstrated by a heat-map and grouped into a *Brassicaceae*-species-specific pathway (IAOX/indole-3-acetadoxime pathway) and a non-*Brassicaceae*-species-specific pathway (including IAM/indole-3-acetamide and IPA/indole-3-pyruvic acid pathway) (Figure 5A). We found that the FPKM values of some synthesis genes in the *Brassicaceae*-species-specific pathway were significantly higher than in the non-*Brassicaceae*-species-specific pathway, especially *BnaC02.NIT2*, *BnaA06.NIT2,* and *BnaC03.NIT2b*, which were homologous of *AtNIT2* (*Nitrilase 2*). These were highly expressed in buds and were only significantly induced in W10 by −B. Interestingly, as a metabolic branch point in the biosynthesis of IAA, two *BnaCYP83B1s* (*BnaA08.CYP83B1* and *BnaC08.CYP83B1*) were significantly inhibited by −B in both bud varieties, which promoted more IAOX to synthetize IAA. Meanwhile, in the non-*Brassicaceae*-species-specific pathway, *BnaA01.TAR2* (*Tryptophan Aminotransferase Related 2*), *BnaC02.TAA1* (*Tryptophan Aminotransferase of Arabidopsis 1*) and *BnaC09.YUC6* were also only induced in W10’s buds by −B. The above results indicated that −B induced the expression of these IAA biosynthesis genes in W10’s buds and contributed to the elevated concentration of IAA.

Moreover, the JA biosynthesis the JA biosynthesis pathway referred to Yan in 2017 [44]. Based on 24 predicted genes involved in the JA biosynthesis pathway in *Arabidopsis*, we blasted 79 genes in *Brassica napus* and separated them into chloroplast-expressed genes (the process of synthesizing OPDA/12-Oxophyto dienoic acid, which is the precursor of JA) and non-chloroplast-expressed (the process of converting OPDA to active JA) genes. Then, 63 genes with the FPKM value were selected for further analysis (Appendix A, Figure 5B). For chloroplast-expressed genes, the expression profiling showed that *BnaFAD3s* (*Fatty Acid Desaturase 3*), *BnaLOX2s* (*Lipoxygenase 2*), and *BnaAOC4s* (*Allene Oxide Cyclase 4*) had higher FPKM values in the transcriptome, which means that these genes play a dominant function in the synthesis of OPDA in ZS11 and W10’s buds. Among them, two *BnaFAD3s* (*BnaA04.FAD3* and *BnaC04.FAD3a*) were significantly suppressed and two *BnaLOX2s* (*BnaC06.LOX2* and *BnaC02.LOX2*) were significantly induced only in W10’s buds by −B. In addition, we also found that the expression of *BnaA05.FAD7b* in W10’s buds was clearly lower than in ZS11’s buds. For non-chloroplast-expressed genes, which were the downstream biosynthetic genes, the heat map presented that *BnaKAT2s* (*3-Ketoacyl-CoA Thiolase 2*), *BnaKAT5s*, *BnaAIM1s* (*Abnormal Inflorescence Meristem 1*) and *BnaMFP2s* (*Multifunctional protein 2*) have higher expression. All the *BnaKAT5s* (*BnaA02.KAT5* and *BnaC02.KAT5*), *BnaC04.KAT2b,* and *BnaA01.AIM1* were significantly inhibited in W10’s buds by −B. Whereas in ZS11, *BnaC02.KAT5* and *BnaC04.KAT2b* presented a down-regulation trend in −B. The decreased expression of these genes caused the reduction in JA in both bud varieties under −B. The above results demonstrated that −B inhibited the expression of these JA biosynthesis genes in both varieties of buds and participated in the reduction in JA, especially in B-inefficient W10.

We also used qPCR to validate the expression patterns of these biosynthesis genes in IAA and JA, and these trends were consistent with transcriptome data (Figure 6). The results showed that these genes indeed responded to −B in the floral buds of rapeseed.

## 3. Discussion

### 3.1. Only the Reproductive Stage Showed a Significant Difference in the Response of Two Contrasting Varieties to Boron Deficiency Under a Pot Experiment Based on Particular Time Point Transplanting

Boron (B) shortage in soils is an important issue in global agricultural production. As one of the world’s major oil crops, the yield and quality of *Brassica napus* are highly dependent on the application of B fertilizer [39,45]. Current research on the regulatory mechanisms of plant response to B deficiency stress has mainly focused on roots and shoots in the vegetative stage, comparing the differences in response to B deficiency in different rapeseed varieties, and also carrying out functional studies on the key genes involved in regulation [26,31,32]. ‘Flowering without seed setting’ is a typical symptom of B deficiency in rapeseed, and has been observed for a long time, but the histological analyses and mechanistic studies of this phenomenon were lacking. This was most likely because B, as a trace element, had a very narrow window between deficient, sufficient, and toxic concentrations in plants, which made it difficult to control rapeseed to respond to B deficiency in the floral organ specifically [46,47]. Verwaaijen et al. carefully induced B deficiency at a particular developmental time point (BBCH15: rapeseed with five true leaves [48]) corresponding to early flowering in a pot experiment, to avoid causing secondary damage to the reproductive stage from B deficiency at the vegetative stage. They revealed that the rapeseed floral organ induced a new pathway and as yet unidentified damage-associated molecular patterns (DAMPs) in response to B deficiency by temporal expression pattern analysis [49]. Similarly to their experimental design, we utilized the B-efficient rapeseed variety ‘Zhongshuang 11’ (ZS11), which had been subjected to high-throughput sequencing [41], and a B-inefficient variety ‘Westar 10’ (W10). We carried out a pot experiment both with and without a B supply, through transplanting similar growth rapeseed varieties at the BBCH15 to observe the differences in the response to B deficiency between the two bud varieties.

In our study, there was no significant phenotypic difference between sufficient (+B) and deficient (−B) B treatments in both rapeseed varieties before bolting (Figure 1A). Entering the flowering stage, the concentration of B in all tissues of W10 significantly decreased under the −B condition, especially in the new leaves and floral organs (Figure 1C). This caused the buds to display symptoms of delayed flowering time and stigma exsertion (Figure 1G and Appendix A), which reduced the siliques length, the number of seeds per siliques, and ultimately, decreased yield (Figure 1H–L). These phenotypic results were consistent with the previous study, indicating that the reproductive stage of W10 was clearly subjected to −B. However, in the case of the ZS11, there were no significant differences in the B concentration of the new leaves and floral organs under the −B treatment, which did not delay flowering time and also had no effect on yield-related indexes (Figure 1C–L and Appendix A). Combined with the phenotypes before bolting, this implied that ZS11 and W10 showed significant differences in response to B deficiency only at the reproductive stage, by transplanting them at the right time point either with or without B supply.

### 3.2. Boron Transporter Genes Differentially Responded to Boron Deficiency in ZS11 and W10 Buds

The uptake and transport of B include three mechanisms: diffusion following the concentration gradient, uptake to the intracellular via channel proteins, and efflux to the extracellular via transporters. Under the −B condition, the plants accomplished B transport mainly through cooperation with channel proteins and transporters [50]. Many members of AQPs (channel proteins) and BORs (transporters) have been reported to respond to −B, and participated in the regulation of B homeostasis [51,52,53,54]. In our study, we identified and characterized 132 AQPs and 21 BORs, and we demonstrated their expression profiles between ZS11 and W10 buds under +B and −B conditions (Figure 3).

In the case of *BnaAQPs*, most of the *BnaPIPs* and *BnaTIPs* genes were relatively higher than others in the buds of both varieties (Figure 3A), which was similar to the transcriptome results in rapeseed roots. *BnaTIP1;1s* and *BnaTIP1;2s* were relatively higher than other *BnaTIPs* genes as well as in the roots. However, *BnaPIP2;4s*, which had higher expression in the roots, had weak expression in the buds, whereas the expression of *BnaPIP1;2s* and *BnaPIP2;7s* was higher. Moreover, *BnaA05.SIP1;1* in the *BnaSIPs* also had a greater expression (Appendix A). It implied that these genes played an important function in the development of buds in rapeseed. Recent studies have revealed that *AtPIP1;2* and *AtSIP1;1* was expressed in the buds, and mainly localized in the papilla cells of pistil, which are involved in the hydration of pollen after pollination [55]. *PIP2;7s*, *TIP1;1s,* and *TIP1;2s* were involved in osmotic regulation in leaves and roots in response to drought and salt stress [56,57,58]. Thus, the novel functions of these genes in the buds remain to be discovered.

Further, the 32 DEGs in *BnaAQPs* with normalized FPKM values and expression patterns were divided into three categories (Figure 3B). Three homologous genes of the *BnaNIP5;1s* and four homologous genes of the *BnaTIP2;1s* were significantly up-regulated mainly in W10 by −B (Appendix A). Previous reports found that *BnaA03.NIP5;1* and *BnaA02.NIP5;1*, like *AtNIP5;1*, were induced by −B in the root, and they were responsible for absorbing B from the environment [25,26]. Both genes were also upregulated in W10 and ZS11’s buds under −B condition, but how they contributed to transporting B in the buds is still unclear. *TIP2;1s* was reported to be involved in water transport in fruits at the reproductive stage, thus affecting their flavor [59,60]. However, it remains to be investigated whether it is involved in bud development or the transport of B. Unlike *AtNIP6*, which was induced by −B and responsible for the distribution of B only in the *Arabidopsis* node, *BnaA02.NIP6;1* (here named *BnaA02.NIP6;1a*) was also expressed in all tissues of the floral organ [28,30]. However, the other three homologous genes of *BnaNIP6;1s* presented an inhibition of expression in W10 by −B. Specifically in *BnaC06.NIP6;1*, the expression decreased by about 20-fold under −B treatment (Appendix A). These results are consistent with the transcriptome of reproductive organs in “*Darmor*” (*Brassica napus* var.) under the −B condition [24]. Therefore, the down-regulation of *BnaNIP6s* could be a key factor contributing to the reduction in B concentration in W10’s buds. Similarly, *BnaA04.SIP2;1* was also inhibited in W10 by −B (Figure 3B). It was found that *AtSIP2;1* would be involved in the regulation of pollen germination and pollen tube elongation, which was critical for the yield [61]. Pollen germination and pollen tube elongation were closely associated with B content [18], hence, whether *BnaA04.SIP2;1* was relevant to the yield decline in W10 under −B requires further research. The third expression trend included *BnaC05.NIP7;1* and *BnaC08. NIP3;1*, which were dramatically up-regulated mainly in ZS11 by −B (Figure 3B, Appendix A). *AtNIP7;1*, a homologous gene of *BnaC05.NIP7;1* in *Arabidopsis*, was involved in transporting B into the anthers and affected pollen development [29]. *BnaC08.NIP3;1* was found to be capable of transporting B in the oocyte system [24]. These two genes may play an important role in enhancing the capacity of B translocation in the buds of B-efficient rapeseed varieties like ZS11. Recent studies also found that *AtNIP2;1*, a closer relative of *AtNIP3;1*, participated in lactate acid efflux under low oxygen stress to reduce cell damage [62]. Thus, whether *BnaC08.NIP3;1* could transport lactate acid, and whether it contributed to the efflux of lactate acid under −B to improve cellular activity, needs to be researched in more detail.

For the *BnaBORs* genes, the B transporters *BnaC4.BOR1;1c* (here named *BnaC04.BOR1a*) [14] and *BnaBOR2s* [31,32], which had been reported to be expressed in floral organs, had relatively higher expression and did not respond to −B in the bud of either variety (Figure 3C, Appendix A). Interestingly, the transcriptional level of *BnaBOR2s* in both buds of the B-efficient variety ZS11 was higher than that of the B-inefficient variety W10, which could improve the capacity of transport B for ZS11 buds. Additionally, the expression of the *BnaBORs* in the bud of both ZS11 and W10 did not respond to −B, except for *BnaA01.BOR7*, which was suppressed by −B only in W10 (Figure 3C, Appendix A). However, current studies about the involvement of *BOR7* in transporting B are lacking.

### 3.3. Differences in the Changes in JA and IAA Concentration in ZS11 and W10’s Buds Under Boron Deficiency

Phytohormones are widely involved in the growth and developmental processes in various plant tissues and play important functions in response to stresses. Presently, many studies focused on the effect of −B on roots mediated by phytohormone signaling [63], but less attention has been paid to the reproductive organs. To define the key phytohormones in response to −B in rapeseed floral organs, we gathered all phytohormone-related genes in the transcriptome, and we spotlighted jasmonate acid (JA) and auxin (IAA) by analyzing DEGs and measuring the hormone concentration of the buds (Appendix A, Figure 4).

JA, as an adversity-responsive phytohormone, was increased in plants subjected to biotic and abiotic stresses, and it improved the resistance by affecting cell wall structure and scavenging reactive oxygen in cells [64,65,66,67]. It was found that −B induced the expression of JA-biosynthesis genes *AtAOCs*, which enhanced the JA concentration in *Arabidopsis* roots and led to inhibition of primary root elongation [36]. However, our results showed that −B significantly reduced the JA concentration in both bud varieties, especially in the B-inefficient variety of W10 (Figure 4E), thus reversing the trend in roots’ response to −B [36,47]. Analyzing the expression profiling of the JA biosynthesis pathway revealed that both *BnaKAT5s* (*BnaA02.KAT5* and *BnaC02.KAT5)*, *BnaC04.KAT2b,* and *BnaA01.AIM*, which were downstream of the biosynthesis pathway and had relatively higher expression, were all suppressed in buds under −B, especially in W10. This could explain the decrease in JA concentration (Figure 4E and Figure 5B, Appendix A). Current research found that pollen development, anther dehiscence, and filament elongation were regulated by JA concentration [68,69], and so a drastic reduction in JA in W10’s buds could severely impair fertility under −B. In addition, the OsMYC2-JA feedback module in rice, which quickly increased the JA concentration in the basal cells of buds, made the cell wall loosen and open flowers [70]. A morphological observation of the buds with time points revealed that the sepals and petals of W10 were more easily abscised compared with ZS11 (Appendix A), which could be attributed to the JA concentration of W10’s buds that was significantly higher than ZS11 (Figure 4E). The delayed flowering time of rapeseed under −B in W10 is probably also closely related to the reduction in JA.

IAA was essential throughout the plant life cycle, and it would make evolutionary sense that plants would share conserved IAA biosynthesis pathway. Two major IAA biosynthesis pathways were tryptophan (Trp)-dependent and Trp-independent. The Trp-dependent pathway could be divided into a *Brassica*-species-specific pathway (IAOX pathway) and a non-*Brassica*-species-specific pathway (including IAM and IPA pathway) [43]. The IAA biosynthesis pathway genes, *TAA1*, *NITs,* and *YUCs*, were up-regulated under −B in the root and the shoot, leading to increased IAA concentrations [71,72]. Under the −B condition, the concentration of IAA was only elevated in the B-inefficient W10 buds and had no variation in the B-efficient ZS11’s buds (Figure 4D). Further analysis of the expression of tryptophan-dependent biosynthetic pathway genes demonstrated that *Brassica*-species-specific pathway genes had relatively higher expression. In particular, *BnaC02.NIT2*, the last step of IAA biosynthesis, was sharply induced by −B in W10. We also found that the expressions of all *BnaCYP83B1s* were suppressed by −B in W10 (Figure 5A, Appendix A). *BnaCYP83B1s* were involved in the biosynthesis of other secondary metabolites from IAOX [73], and thus the down-regulation of *BnaCYP83B1s* allowed for more IAOX to synthesize IAA, and thus increase the IAA concentration, although this hypothesis needs further research. In the non-*Brassica*-species-specific pathway, we found that *BnaA01.TAR2*, *BnaC02.TAA1,* and *BnaC09.YUC6* were significantly induced by −B only in W10’s buds, and also increased the IAA concentration (Figure 4D, Appendix A). The excessive accumulation of IAA led to anthers indehiscence, pollen grains inactivation, and ultimately, resulted in sterility [74]. Additionally, there was an uneven distribution of IAA-mediated root bending and leaf curling [75,76]. We hypothesized that the stigma bending of W10 under −B could be the result of the uneven distribution of IAA, which also requires further investigation (Appendix A).

In summary, by analyzing the expression pattern of phytohormone-related genes and determining the phytohormone concentration in the buds of the two varieties under +B and −B treatments, we found that the concentration of JA decreased in both ZS11 and W10’s buds under −B. We also found that the concentration of IAA increased only in the B-inefficient W10 buds (Figure 4D,E). By integrating the analysis of gene expression on the biosynthetic pathway, we suggest that *BnaC02.NIT2*, together with *BnaKAT5s*, could be the key genes leading to the variation in IAA and JA in both varieties under −B conditions (Figure 5 and Figure 6).

## 4. Materials and Methods

### 4.1. Plant Materials of Brassica napus and Cultivation Conditions

Two different *Brassica napus* varieties, B-efficient “Zhongshuang 11” (ZS11) and B-inefficient “Westar 10” (W10) [34,41], were used to conduct the pot experiment. All rapeseeds were first cultivated in Hoagland solution (containing 5 mM KNO_3_, 5 mM Ca(NO_3_)_2_, 1 mM KH_2_PO_4_, 2 mM MgSO_4_, 0.05 μM Fe-EDTA, 0.3 μM CuSO_4_, 0.8 μM ZnSO_4_, 0.01 μM MnCl_2_, 0.4 μM Na_2_MoO_4_) with 46 μM B [77] for 25 d until rapeseed had about 5 true leaves (BBCH15). Homogenous plants were washed with ultra-pure water (18.25 MΩ·cm) to remove possible apoplast B, and then the plants were randomly planted into the prepared pots under B supply (+B: 1.0 mg/kg B) or free (−B: 0.24 mg/kg B) conditions with 4 independent replications. The potted soil was sandy loam, which was obtained from Guotan village, Meichuan town, Hubei province, China (30°17′ N, 115°59′ E). The basic physicochemical properties of the soil included pH 5.32, organic matter 30.67 g/kg, alkali-hydrolyzed nitrogen 104.1 mg/kg, available phosphorus 10.14 mg/kg, available potassium 145.0 mg/kg, and available B 0.24 mg/kg. Each pot contained 7 kg of air-dried and sieved soil. The ultrapure water (18.25 MΩ·cm) was used to irrigate throughout the whole growth stage, and the irrigation frequency was 1–2 times per day according to the demand of plant growth. When the two varieties of rapeseed grew into the flowering stage, the early closed buds with unified similar sizes (buds with 2–4 mm length) were harvested from the main inflorescence under +B or −B conditions. The samples of each treatment included 6 biological replications with a fresh weight of about 100 mg for each replication, and were stored in a 2 mL centrifuge tube with free RNAase. Then, they were transferred to −80 °C for storage. Among the 6 biological replications, 3 biological replications were randomly selected for transcriptome sequencing and the other 3 biological replications were used for phytohormone measurement.

### 4.2. Measurement of Boron Concentration in Pre-Bolting and Flowering Stage by ICP-OES

Different tissue parts of ZS11 and W10 before bolting (root, old leaves (1st and 2nd leaves), with new leaves (8th and 9th leaves)), at flowering stage (stem, new leaves (10th and 11th leaves), or with old leaves (3rd and 4th leaves), floral organ), were harvested and washed three times with ultra-pure water (18.25 MΩ·cm). All samples were dried in the oven at 65 °C for about 2 days until the dry weight stabilized to a constant weight. The dried samples were ground into fine powder using a carnelian mortar. Fine powder was transferred into 50 mL centrifuge tubes and 10 mL of 1 mol/L HCl (UR) solution was added. Then, the mixture in the tubes was shaken at 200 rpm for 3 h. After the filtration of the solution, the B concentration was measured by ICP-OES analysis (Thermo Scientific iCAP 6000 Series, Cambridge, UK).

### 4.3. Morphological Changes with Time in Floral Organ

When both ZS11 and W10 reached the flowering stage under +B and −B treatments, one rapeseed per pot was selected, and all buds which had just formed at the top of the main stem were marked with a pen. Every 2 days, the marked buds were harvested into a 50 mL centrifuge tube, capped tightly, and immediately placed in a −20 refrigerator until the petals of all the marked buds had withered. All samples were carefully removed from the refrigerator and lightly covered with absorbent papers at room temperature before being photographed.

### 4.4. Analysis of the Transcriptome Approach

For the transcriptome analysis, the total RNA was isolated using the Trizol Reagent (Invitrogen Life Technologies, Carlsbad, CA, USA). The quality and integrity of the RNA were determined using a NanoPhotometer^®^ spectrophotometer (IMPLEN, Village, CA, USA) and an RNA Nano 6000 Assay Kit of the Bioanalyzer 2100 system (Agilent Technologies, Santa Clara, CA, USA). A total amount of 1 µg RNA per sample was used as input material for the RNA sample preparations. Sequencing libraries were generated using NEBNext^®^ UltraTM RNA Library Prep Kit for Illumina^®^ (NEB, USA). The brief steps were as follows: mRNA was purified from total RNA using poly-T oligo-attached magnetic beads. Fragmentation was carried out using divalent cations under elevated temperature in NEBNext First Strand Synthesis Reaction Buffer(5X). First-strand cDNA was synthesized using a random hexamer primer and M-MuLV Reverse Transcriptase (RNase). Second-strand cDNA synthesis was subsequently performed using DNA Polymerase I and RNase H. The library fragments were purified with the AMPure XP system (Beckman Coulter, Beverly, MA, USA). Then, 3 µL IUSER Enzyme (NEB, Ipswich, MA, USA) was used with size-selected, adaptor-ligated cDNA at 37 °C for 15 min followed by 5 min at 95 °C before PCR. Then, PCR was performed with Phusion High-Fidelity DNA polymerase, Universal PCR primers, and Index (X) Primer. At last, PCR products were purified (AMPure XP system) and library quality was assessed on the Agilent Bioanalyzer 2100 system. The clustering of the index-coded samples was performed on a cBot Cluster Generation System using TruSeq PE Cluster Kit v3-cBot-HS (Illumia, San Diego, CA, USA). After cluster generation, the library preparations were sequenced on an Illumina Novaseq platform, and all reads were analyzed by Beijing Novegene Co., Ltd. (Beijing, China). Raw data (raw reads) of the FASTA format were first processed through in-house perl scripts. The reference genome and gene model annotation files were downloaded from the genome website (http://brassicadb.cn/#/download/, accessed on 21 May 2023) directly. FeatureCounts v1.5.0-p3 was used to count the reads numbers mapped to each gene. Then, the FPKM of each gene was calculated based on the length of the gene and the reads count mapped to this gene. The differential expression analysis of two conditions was performed using the DESeq2R package (1.16.1). The resulting *p*-values were adjusted using Benjamini and Hochberg’s approach for controlling the false discovery rate. Genes with an adjusted *p*-value ≤ 0.05 and an absolute value of log_2_Fold-change ≥ 1 as the standards to define differentially expressed genes (DEGs). Gene Ontology (GO) enrichment analysis of DEGs was implemented by the clusterProfiler R 4.1.1. package.

### 4.5. Phytohormone Measurement

Methods for the measurements of IAA, JA, and ABA referred to Zhou [34]. Samples with recorded fresh weights in 2 mL centrifuge tube were first ground into fine powder (50 Hz, 1 min), and 750 μL Buffer I (methanol: ultrapure water: acetic acid = 80:19:1, *v*/*v*/*v*) with internal standards (contained 30 ng ^2^H_6_ABA (OlChemIm, Olomouc, Czech Republic), 40 ng DHJA (OlChemIm), and 50 ng D_2_-IAA (Sigma-Aldrich, St. Louis, MI, USA)) were added to each sample. After shaking (300 rpm) for 16 h in the dark at 4 °C, the mixture was centrifuged for 10 min (4 °C, 12,000 rpm) and the supernatant was transferred in a new centrifuge tube. Next, 450 μL Buffer II (methanol: ultrapure water: acetic acid = 80:19:1, *v*/*v*/*v*) was added to the sediment. After shaking (300 rpm) for 4 h in the dark at 4 °C and centrifuging for 10 min (4 °C, 12,000 rpm) again, the two supernatants were then combined and filtered through a 0.22 μm filter into a new centrifuge tube. Volatilization was carried out by high-speed centrifugation, and 200 μL 50% methanol was added to dissolve. Finally, the mixture was centrifuged at 12,000 rpm at 4 °C for 15 min, and the supernatant was transferred to the sample tube. The concentrations of IAA, JA, and ABA were determined by ultra-fast liquid chromatography-electrospray ionization tandem mass spectrometry (UFLC-ESI-MS).

### 4.6. RNA Extraction, Reverse Transcription, and Quantitative Real-Time PCR (qRT-PCR)

Total RNA was extracted independently using Trizol reagent (TRIpure Reagent, Aidlab, Beijing, China). Then, the concentration and absorbance of the total RNA were measured by UV spectrophotometer (NanoDrop 2000c, Thermo Scientific, Waltham, MA, USA). For each sample, cDNA was synthesized based on 2000 ng of total RNA by the Reverse Transcription kit (TRUEscript RT Kit, Aidlab, Beijing, China). For quantitative real-time PCR, 2x Universal SYBR Green Fast qPCR Mix (ABclonal, Wuhan, China) was applied to the CFX96TM Real-time PCR Detection System (Bio-Rad, Hercules, CA, USA). There were four biological samples for each tissue and three technical replicates for each sample. *Brassica napus EF1-α* and *Tublin* were used as reference genes, and all primers were listed in Appendix A. The gene expression was calculated according to the 2^−ΔΔCt^ method.

### 4.7. Data Analysis

All statistical data in this study were analyzed with Student’s *t*-test (* *p* < 0.05, ** *p* < 0.01, *** *p*< 0.001) by using the Prism 8.0 software (GraphPad Software, La Jolla, CA, USA).

## 5. Conclusions

Pot experiments based on developmental time point transplanting led to significant differences in the response of two plant varieties to −B only at the reproductive stage. The W10’s floral organs were more vulnerable to −B stress. Transcriptome analysis found that *BnaC05.NIP7;1*, *BnaC08.NIP3;1,* and *BnaBOR2s* were identified as the key genes which could enhance the capacity of B translocation in buds of ZS11 and *BnaC02.NIT2*, which, together with *BnaKAT5s*, could be the key genes responsible for the changes in IAA and JA concentrations in W10’s buds, which likely resulted in the higher sensitivity of W10’s buds to −B.

## Figures and Tables

**Figure 1 plants-14-00859-f001:**
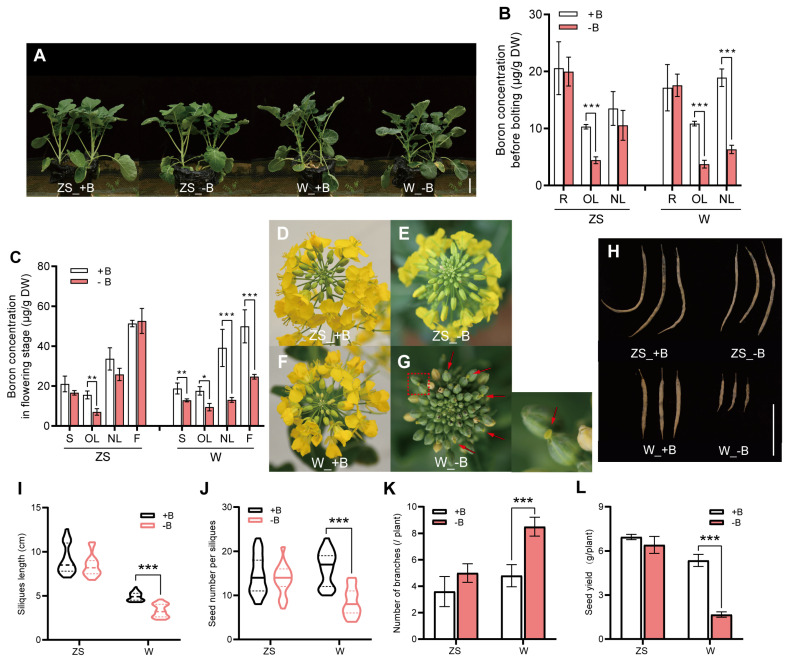
Pot experiment assessing the phenotypical differences between ZS11 and W10 varieties in response to boron (B) deficiency. *Brassica napus* cultivars ZS11 (B-efficient) and W10 (B-inefficient) were transplanted into a pot with a particular time point (BBCH15) under a sufficient (+B) or deficient (-B) boron supply. The phenotype (**A**) and B concentration (**B**) of the root, old leaves, and young leaves before bolting. (**C**) B concentration of stem, old leaves, new leaves, and floral organ. The distinct phenotype in inflorescence (**D**–**G**) and siliques (**H**) between W10 and ZS11 at the reproductive stage under +B or −B. (**I**) Silique length, (**J**) seed number per silique, (**K**) number of branches, and (**L**) seed yield between two genotypes at the reproductive stage under +B or −B. R: root, OL: old leaves, NL: new leaves, S: stem, F: floral organ. Values represent mean ± SD. *n* = 4 (**B**,**C**), *n* = 30 (**I**,**J**) or *n* = 3 (**K**,**L**) independent replicates, *t*-test, * *p* < 0.05, ** *p* < 0.01, *** *p* < 0.001. Bar = 10 cm (**A**) or 5 cm (**H**). The red arrows (**G**) mark the buds with stigma exsertion.

**Figure 2 plants-14-00859-f002:**
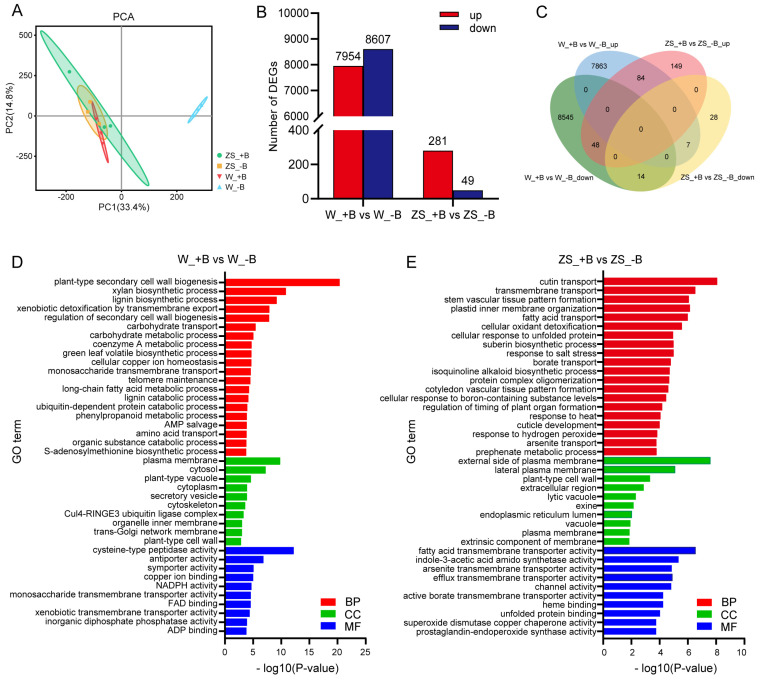
Characterization of the differentially expressed genes (DEGs) in ZS11 or W10’s floral buds under sufficient or deficient boron supplies. (**A**) Principal component analysis (PCA) for the transcriptome of ZS11 or W10 buds harvested under +B or −B conditions. (**B**) Number of the up-regulated and down-regulated DEGs in ZS11 or W10 with different boron supplies. (**C**) Venn diagram showing the distribution of DEGs regulated by different boron conditions between ZS11 and W10 buds. Gene ontology (GO) enrichment analysis of DEGs in W10 (**D**) or ZS11 (**E**) buds under +B and −B conditions. ZS11 or W10 buds were harvested with a unified similar size from +B or −B treatment for transcriptome analysis. +B: Sufficient boron condition, −B: deficient boron condition. ZS: Zhongshuang 11, W: Westar 10. BP: biological process, CC: cellular component, MF: molecular function. The top 20 (BP) or 10 (CC/MF) terms of each category were listed. The definition of DEGs: *p*_adj_ value ≤ 0.05 and the absolute value of log_2_(fold-change) ≥ 1.

**Figure 3 plants-14-00859-f003:**
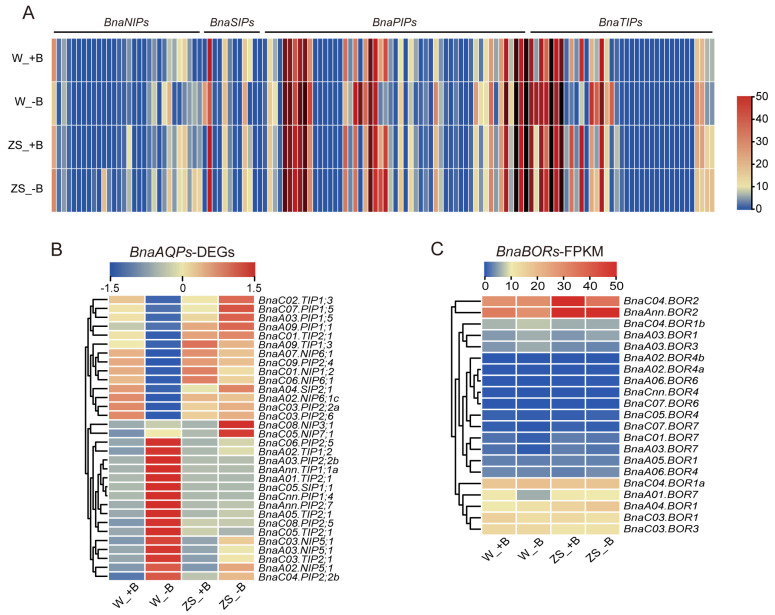
Heat map showing the expression profiling of BnaAQPs and BnaBORs family members in ZS11 or W10 buds under sufficient or deficient boron conditions. (**A**) The general expression profiling of *BnaAQPs*. (**B**) Relative expression of DEGs in *BnaAQPs* by normalized FPKM values. (**C**) The general expression profiling of *BnaBORs*. W10 and ZS11 buds were harvested with unified similar sizes from +B or −B treatment. +B: Sufficient boron condition, −B: deficient boron condition. ZS: Zhongshuang 11, W: Westar 10. The criterion for the identification of DEGs was as follows: *p*_adj_ value ≤ 0.05 and the absolute value of log_2_(fold-change) ≥ 1. Color scales indicate the values of FPKM values (**A**,**C**) or normalized FPKM values (**B**). Red indicates high expression, white indicates intermediate expression and blue indicates low expression.

**Figure 4 plants-14-00859-f004:**
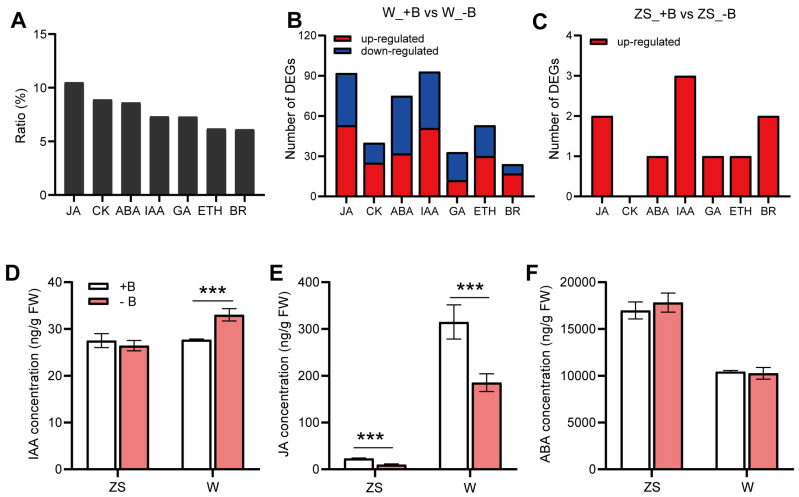
Differentially expressed phytohormone-related genes in transcriptome and the phytohormone concentrations. (**A**) The ratio of DEGs of each phytohormone in response to −B in transcriptome to all related genes for the corresponding phytohormone. (**B**) The number of differentially expressed various phytohormone-related genes in response to −B in W10. (**C**) The number of differentially expressed various phytohormone-related genes in response to −B in ZS11. The concentration of IAA (**D**), JA (**E**), and ABA (**F**) in both ZS11 and W10’s buds under different B conditions. Values represent mean ± SD. *n* = 3 (**D**–**F**) independent replicates, *t*-test, *** *p* < 0.001.

**Figure 5 plants-14-00859-f005:**
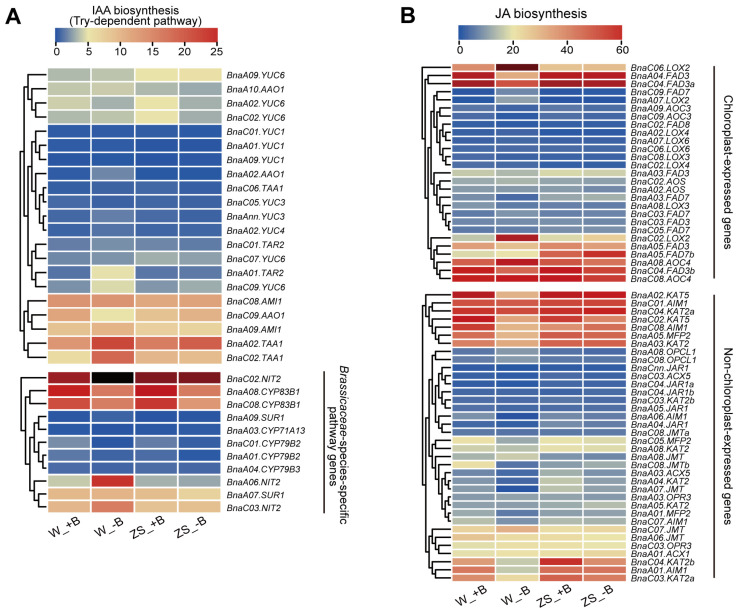
Heat map showing the expression profiling of IAA and JA biosynthesis genes in ZS11 or W10 buds under sufficient or deficient boron conditions. The general expression pattern of IAA (**A**) and JA (**B**) biosynthesis genes in both ZS11 and W10 buds under different B treatments. Color scales indicate the values of FPKM values. Red, white, and blue indicate high, intermediate, and low expression, respectively.

**Figure 6 plants-14-00859-f006:**
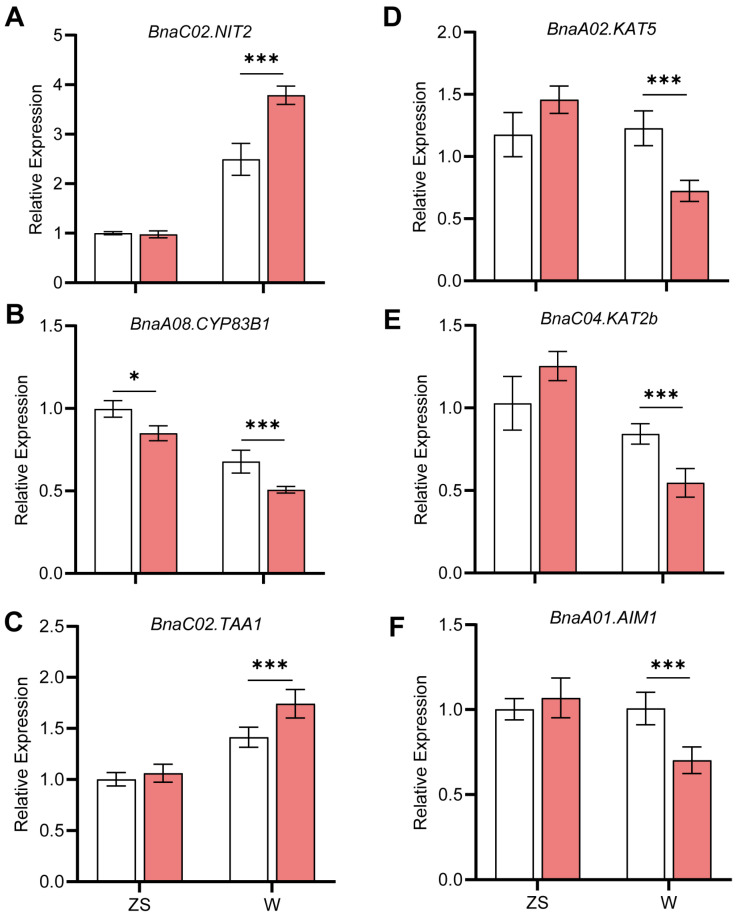
qPCR analysis of expression patterns. Confirming the expression profile of genes on the IAA (**A**–**C**) or JA (**D**–**F**) biosynthesis pathway in the transcriptome. Values represent mean ± SD. *n* = 4 independent replicates, *t*-test, * *p* < 0.05, *** *p* < 0.001.

## Data Availability

The original contributions presented in this study are included in the article/Appendix A. Further inquiries can be directed to the corresponding author.

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
