# Peer review of "Transcriptional Analysis Reveals the Differences in Response of Floral Buds to Boron Deficiency Between Two Contrasting Brassica napus Varieties"

_plants, 2025, doi:10.3390/plants14060859_

Round 1
Reviewer 1 Report
Comments and Suggestions for Authors
This study investigates the response of reproductive organs to boron (B) deficiency in Brassica napus. Two varieties were compared under B-sufficient and B-deficient conditions. The authors identified phenotypic differences, conducted transcriptome analysis, and pinpointed key genes and phytohormone changes associated with B deficiency in the reproductive stage. The findings provide insights into B-efficient utilization in rapeseed and its potential to improve yield.In conclusion, the paper would be sufficient to merit publication in Plants, though a minor revision is recommended, which needs to include the following points.
(1) Ensure all figures and tables are clear. Some figures are too small. The description of the phenotypic responses (e.g., flower opening, stigma exsertion) is detailed but could be supported with more high-quality images or graphs for clarity.
(2) The abstract lacks a statement of broader implications. Adding a concluding sentence on how these findings could influence agricultural practices would improve it
(3) Add “reproductive development” to the keywords for broader visibility.
(4) Line 13: typic B-deficient -> “typical B-deficient
(5) Line 28: could predominantly led to -> could predominantly lead to
(6) Line 351: was significantly decreased -> significantly decreased
(7) Line 504: Clarify if boron was measured across multiple time points or only at
specific stages (e.g., pre-bolting vs. flowering).
(8) The sampling strategy and number of biological replicates for RNA-seq and phytohorumone measurement are not well described. Include details on replication strategy for RNA-seq and phytohormone assays to ensure reproducibility.
(9) In Figure 2, the PCA analysis includes ellipses around each group of samples, but they
appear arbitrary. Please remove the ellipses and show only the points.
Comments on the Quality of English Language
There are some grammatical mistakes, so it would be good to have it checked by a native speaker.
Author Response
Sincerely thanks for your review and comments. We have provided a point-to-point response in the Word file and please see the details in the attachment.

Reviewer 2 Report
Comments and Suggestions for Authors
The study investigates the response of two Brassica napus cultivars, ZS11 and W10, to boron (B) deficiency, focusing on the reproductive stage. ZS11, a B-efficient variety, showed fewer symptoms of B deficiency compared to W10, which showed delayed flower opening and reproductive failure. Transcriptome analysis revealed that W10 had more differentially expressed genes (DEGs) than ZS11, indicating greater susceptibility to B deficiency. The manuscript contains a large amount of data, mainly at the genetic level, but I have some concerns about the experimental design of the manuscript.
The abstract is quite good and explains the basis of the manuscript and the work done by the authors. I found that the last sentences of the abstract are not very clear. For example, some genes are mentioned, but their function or the function of the protein they encode is not specified. So, readers who are not familiar with these genes may be a little confused by the gene names. Please add more details about the genes BnaC02.NIT2 and BnaKAT5s.
Another point that is not well defined in the abstract is the relationship between boron and phytohormones. The authors mention that they analyzed the levels of phytohormones, but it is not specified why. In general, the abstract lacks a rationale for the authors' experimental approach.
Please revise the second paragraph of the introduction. In the first paragraph, you did an excellent job of explaining why boron is important in plants, especially in the formation of reproductive organs. The first paragraph of your introduction ends with a question: There is little information about the molecular mechanism of reproductive abortion following boron deficiency. In the second paragraph, you are expected to outline your experimental approach to solving this problem. In fact, you mentioned several genes and proteins that could facilitate boron uptake. So, I ask you to rewrite the second paragraph to explain how you wanted to approach the problem mentioned in the first paragraph.
In the third paragraph, you introduced the rapeseed, which is the plant in which you studied the boron deficiency. That's fine, but I'd like you to tie the description of this plant more closely to what you said in the other two introductory paragraphs.
In general, the introduction lacks a link between boron transporters and phytohormones, which are the two main topics of your work. So, you need to define a stronger connection between these two topics.
In addition, since both cultivars have already been described and partially characterized, you will need to emphasize why and how your work differs from previous work. I mean, since some data has already been provided for both cultivars, please outline what is new in this manuscript compared to other papers.
Please use standard nomenclature in figures. I mean, if you use ZS11 in some figures, please use that name in all figures. I see that in some graphs only ZS is displayed. The same is true for W10.
Transcriptome data is a good example of a lack of description of the experimental approach. In Figure 1 you described that boron is distributed differently between the two varieties, I mean with the more boron efficient variety being able to distribute boron into new leaves and floral organs while the boron inefficient variety is not able to do the same. However, when you approached transcriptome analysis, you isolated RNA from buds. My question is: What is the relationship between the analysis of boron in new leaves, old leaves, stems and flower organs and then the transcriptome analysis in buds? What did you expect to find by analyzing the expression in buds in relation to the altered boron distribution between old and new leaves? What is the rationale for this approach? This is a very critical point.
The choice to analyze gene expression in buds may be logical, but some questions may arise. For example, buds usually go through different stages of development, and I'm not sure that the expression of genes is always the same during all stages of bud development. So how did you choose the stage of bud development at which you analyzed gene expression? How can you be sure that gene expression in buds reflects boron accumulation in fully developed flowers?
Another point that deserves attention is what I described earlier, namely the link between phytohormones and boron accumulation. The connection between these two aspects is not fully described in the introduction, so now it is not easy to follow the experimental approach and results described by the authors.
Back to phytohormones. I noticed that the authors measured phytohormone levels in buds as well as gene expression. So again, the authors focused and limited their attention to the buds. My general question is why focus on buds only? I think that the distribution of boron may be due not only to the activity of the flower buds, but also to how much boron is released from the leaves. Why not study the phytohormones in different organs of plants? I honestly do not understand the experimental approach of the authors. They talk about boron in the plant with a focus on floral organs, and then they focused their analysis only on buds. I'm not sure that what they found can give an accurate picture of how boron is distributed in rapeseed plants and specifically in flower organs.
Comments on the Quality of English LanguageNothing of note to report
Author Response

(The authors gave the same response as above.)

Reviewer 3 Report
Comments and Suggestions for Authors
Jiang et al. provide valuable in-depth insights for boron deficiency among two physiologically contrasting varieties of Brassica napus. This piece of work consists a strong, solid one step further continuity of previously published data for B. napus boron’s deficiency, mainly from the same research group [https://doi.org/10.1016/j.ecoenv.2024.116011 ] Both examined varieties attract universal interest for Boron studies [https://doi.org/10.1111/tpj.13669 ] due to available in silico and wet lab genetic information. The manuscript is title-focused, well-structured, well-written and it covers all potential aspects of boron deficiency for the studied B. napus varieties. Techniques were set in order to provide coherent series of data which end up to solid conclusions; experimental design covers all topic aspects from macro to micro scale extremely sufficiently. Scientific references support and explain clearly the presented data. No major/minor weaknesses were found on the text and supplemented material; I would suggest that it can be published as it stands.
Author Response
Sincerely thanks for your reviewing the article and recognizing our research

Round 2
Reviewer 2 Report
Comments and Suggestions for Authors
I decided to accept this manuscript, but I believe that the study's focus on the buds for both genetic and phytohormone analysis may prevent the paper from being referenced in future studies. The manuscript is correct in the experimental section but falls short in the experimental design.